# Management of Bilateral Quadriceps Tendon Ruptures Post Total Knee Arthroplasty by Kesler Technique Using Fiber Tape

**DOI:** 10.3390/healthcare11050631

**Published:** 2023-02-21

**Authors:** Waleed Ahmad AlShaafi, Mohammed Hassan Alqahtani, Abdullah Hassan Assiri, Abdulrhman Abdullah Alqhtani, Jaya Shanker Tedla, Dhuha Saeed Motlag

**Affiliations:** 1Consultant Orthopedics and Arthroplasty, Saudi German Hospital-Aseer, Abha 61411, Saudi Arabia; 2Department of Orthopedics, Aseer Central Hospital, Abha 61411, Saudi Arabia; 3Department of Medical Rehabilitation Science, College of Applied Medical Sciences, King Khalid University, Abha 61421, Saudi Arabia

**Keywords:** total knee arthroplasty, quadriceps tendon ruptures, surgery, graft, Kesler technique, fiber tape

## Abstract

Total knee arthroplasty is an effective way to manage osteoarthritis patients surgically. However, patients may encounter post-surgical complications, such as quadriceps rupture in rare instances, in addition to surgical complications. In our clinical practice, we encountered a 67-year-old Saudi male patient with a rare bilateral quadriceps rupture two weeks post-total knee arthroplasty. The cause of the bilateral rupture was a history of falls in both knees. The patient was reported to our clinic with clinical features like pain in the knee joint, immobility, and bilateral swelling in the knees. The X-ray did not show any periprosthetic fracture, but an ultrasound of the anterior thigh revealed a complete cut of the quadriceps tendon on both sides. The repair of the bilateral quadriceps tendon was done by direct repair using the Kessler technique and was reinforced with fiber tape. Following knee immobilization for six weeks, the patient began intensive physical therapy management to decrease pain, enhance muscle strength, and increase range of motion. After rehabilitation, the patient regained a complete range of motion in the knee and improved functionality, and he could walk independently without crutches.

## 1. Introduction

Osteoarthritis is the most common degenerative disease-causing articular dysfunction; the accompanying pain and limited range of motion can eventually lead to decreased functional capacity and reduced quality of life [1,2]. Osteoarthritis is one of the most challenging disorders for the orthopedic rehabilitation team. Due to the lack of any disease-modifying medications, severe pain, limitation in the range of motion, difficulty in walking, and inability to perform day-to-day activities, most patients will ultimately undergo surgery [3]. One of the most effective surgical options for patients with severe knee osteoarthritis is total knee arthroplasty [2,3]. Postoperatively, after total knee arthroplasty, there are some common complications, such as infection, venous thromboembolism, the persistence of pain, and mortality [4,5,6]. However, rupture of the bilateral quadriceps post total knee arthroplasty is a rare complication, but it leads to serious functional limitations [7]. The knee’s extensor mechanism involving the quadriceps muscle is crucial for locomotion. The four muscles of the quadriceps (namely, the vastus medialis, vastus lateralis, vastus intermedius, and rectus femoris), tendon of the quadriceps, patella, patellar tendon attachment to tibia, and retinaculum are the critical components of knee extension [6,7]. A concurrent, unexpected quadriceps tendon rupture on both sides after total knee arthroplasty is an unlikely orthopedic injury [8,9]. A disturbance in the extensor mechanism is a severe consequence of total knee arthroplasty, with a reported incidence of 0.17 to 2.5 percent [10,11]. A disruption of the quadriceps tendon is commonly related to the failure of surgical technique or patient-related factors [11]. A dysfunction in the extensor mechanism post-total knee arthroplasty may be caused by numerous factors ranging from medically induced reasons to post-traumatic disturbance. Such dysfunction in the knee extensor mechanism ranges across acute, subacute, and chronic stages [12]. Several surgical options have been proposed for quadriceps repair. The gap between the incised quadriceps muscle, the patient’s general demographic characteristics, the duration of injury, and tissue pathophysiology all help to inform the type of surgery required. In some cases, direct repair by getting the two ends of the muscle together with sutures will be sufficient, whereas in other cases there may be a need to use an external material other than the same muscle; among alternative materials, the most common are allografts where the muscle graft is obtained from another human, an autograft where the graft is obtained from other muscles in the same human, and synthetic mesh, which is an artificial material used to reinforce the surgical repair [13,14,15,16]. 

In this case study, we describe a case presented in our clinical setting two weeks after total knee arthroplasty; we saw a 67-year-old Saudi man with a rare bilateral quadriceps rupture. The bilateral rupture was provoked by a history of falling on both knees. The bilateral quadriceps tendons were repaired directly using the Kessler approach and strengthened with fiber tape. In the Kessler repair, a knot is used to secure the suture to the tendon at each of its four corners. This knot stops the suture from slipping inside the tendon material [17]. The Kessler suture procedure is simple and time efficient, which is an essential feature during surgery [18]. The Kessler approach using fiber wire suture significantly enhances the mechanical qualities of intra-synovial tendon repair and lowers the possibility of postoperative gapping and rupture [19]. The objective of our case study is to provide more details regarding the case in terms of the patient’s mechanism of injury, area of the damage in the muscle, diagnosis procedures used, surgical procedure, and the patient’s recovery journey after rehabilitation. 

## 2. Case Description

### 2.1. Pre-Intervention Phase

A 67-year-old Saudi male patient visited our outpatient department with knee joint pain on both sides and a diagnosis of bilateral knee osteoarthritis. His preoperative examinations were standard except for a deficiency in vitamin D, which was corrected by oral supplement. The patient was a known diabetic and hypertensive but had no history of steroid use or diagnosis of rheumatoid arthritis. His bilateral knee joint X-rays revealed the severity of the disease and osteoarthritic changes, as shown in Figure 1.

Subsequently, the patient underwent total bilateral knee arthroplasty by the standard midline approach on the para patellar region by Zimmer, USA, without lateral retinacular release. The patient’s surgery was successful; his postoperative knee joint X-rays can be seen in Figure 2.

The patient revisited our clinic two weeks after the surgery with pain in both knees and the inability to walk following a fall on his operated knees. A physical assessment showed swelling on both knees, no gross joint deformity, a bilateral supra-lateral gap, and tenderness. Although a clinical examination of the knee joint failed to show the extensor mechanism, it was distrusted by a positive leg-raising test that was conducted bilaterally, as shown in Figure 3. 

Radiological screening with a bilateral knee X-ray showed no periprosthetic fracture, as seen in Figure 4. 

A lateral view of the knee joint X-rays also revealed no acute fracture but showed a lower positioning of the patella, known as patella baja. An ultrasound report of the quadriceps disclosed a complete bilateral rupture of the quadriceps tendon (Figure 5). On the ultrasound, the gap between two ruptured segments of the quadriceps was less; however, intraoperatively, when we observed the gap between the segments, it was larger, with a 1.4 cm defect in the right quadriceps and a 2.4 cm defect on the left side.

### 2.2. Intervention Phase

The subject underwent surgery to repair the disrupted quadriceps. Anesthesia was administered using the spinal route, and we started surgery with prophylactic antibiotic medications and applied a tourniquet bilaterally. Using the sterile technique of prepping and draping at the surgical site, we identified the previous midline incision site and extended the incision proximal to the previous one in order to identify the quadriceps rupture. During the procedure, the surgeons observed a mid-substance complete tear of the right quadriceps and a distance complete tear of the left quadriceps, as shown in Figure 6. 

However, the left quadriceps tear was distal compared with the right side. Commonly in these situations, we use Ethibond for suturing and repairing purposes; however, upon considering the patient’s characteristics, such as age, weight, and muscle weakness, we intended to secure it better. Hence, the surgeons decided to use an innovative method where the primary direct repair of the bilateral quadriceps tendon was done utilizing the Kessler technique using fiber tape (Braided Polyblend Suture) provided by Arthrex^®^. The intraoperative post-suture quadriceps muscle can be seen in Figure 7. 

### 2.3. Post-Intervention Phase and the Results

The patient was on a knee immobilizer for six weeks, and during immobilization, isometric exercises for quadriceps muscle were encouraged. Post immobilization, the patient came to the outpatient department for follow-up and was referred for regular physical therapy sessions. After six weeks, he began pain management using transcutaneous electrical stimulation. In addition, muscle strengthening was done using a quadriceps table in the therapy sessions and using weight cuffs at home in the available ranges. The muscle strength was measured using a Baseline Hydraulic Push–Pull Dynamometer of 250 pounds with a dial (analog) gauge (Model: FEI-12-0394, Fabrication Enterprises Inc., USA). The maximum resistance that the patient could exert was assessed; this resistance measured in pounds is called a one-repetition maximum. The forty percent load of this one repetition maximum was used as the training weight initially. In our case, in the sixth week, the patient could exert around eight pounds as the one repetition maximum, and his training weight was approximately three pounds. This training intensity was progressively increased in the later stages by increasing the weight and repetitions. More details of the load percentage progression, number of repetitions per set, number of sets per session, and number of sessions per week are described in Table 1.

Gradual range of motion exercises were performed with knee flexion 0–30, 0–60, 0–90, and 0–120 degrees, followed by full flexion. Along with these open kinematic strengthening exercises, we performed close kinematic eccentric muscle strengthening by quadriceps lunge exercises. Finally, the patient obtained a good range of motion three months post-repair, with excellent results in function. His straight leg-raising test [20] can be seen in Figure 8 for both legs (Appendix A). Before sending this research study for publication, the authors obtained written informed consent from the patient, and his identification and medical details were maintained confidentially.

## 3. Discussion

Quadriceps rupture following a complete knee replacement is an uncommon but catastrophic impediment that can develop in any patient [21,22,23,24]. The quadriceps muscle is an essential muscle for maintaining a standing posture. When a complete rupture happens bilaterally, it is devastating for patients because it is so painful that patients cannot sit, stand, walk, reach, or bend. Unless the damage is repaired bilaterally, the subjects cannot do their activities of daily living and participate in any of their life-related skills. After our study’s surgical repair of the quadriceps tendon, we obtained an excellent outcome with an improved range of motion and function after six months. Many individuals with complete extension mechanism disruption have required surgical intervention, with varied results [22]. Upon examining the literature, we discovered that quadriceps tendon repair generally has a good outcome [25,26]. Primary repair of acute quadriceps rupture at the tendon level is understood as repair with sufficient tissue within two weeks of injury. This type of repair has the best results and has only a two percent reoperation rate [26,27]. The treatment for complete disruptions is determined by the damage’s place, degree, and duration, the subjects’ overall health and expectations, and the performing surgeon’s proficiency and capability [14]. Principle repair using cerclage wire or suture augmentation, followed by restriction of the knee’s range of movement in full extension, is the most frequently described procedure [15,16]. 

Ormaza et al. conducted a case series on managing chronic quadriceps rupture post-total knee arthroplasty using synthetic mesh. They had three subjects who had chronic partial unilateral quadriceps tears and underwent surgical repair using a suture reinforced with synthetic mesh after the failure of conservative management. The patients improved significantly after this procedure [7]. In our case, the quadriceps injury was unique because it was an acute injury that happened within two weeks of total knee arthroplasty; it happened bilaterally. In addition, the primary direct repair was done by the unique Kessler technique of fiber tape suturing. Even though there are several surgical options available, like allograft, autograft, and synthetic mesh, for repairing quadriceps rupture, the surgeons in the current study had considered Kessler’s technique because of the following reasons. First, the extent of the injury was less than 5 cm; if there is a more significant difference between two ends of the ruptured quadriceps muscle, then it is challenging to approximate them and perform the direct primary repair without the use of any external materials. Second, the injury was acute within two weeks after total knee replacement, it was bilateral, and the patient was old and already had a bilateral total knee replacement, hence the outlook for performing autograft or allograft and having a subsequent favorable level of recovery was very poor [26,27]. Since there is no bone fragment (such as, for example, in patellar tendon ruptures), the use of Illizarov surgery is also not possible because the ruptured segments are soft tissues [28].

Post-repair outcomes are linked to the period between injury and repair; patients with immediate primary repair are more likely to experience a favorable functional outcome, full range of motion, and strong quadriceps. We believe that in our case, the immediate direct repair using the Kessler technique with reinforced fiber tape, along with good rehabilitation and patient factors, made it possible for excellent improvements in the patient’s recovery of function. 

Like all other case studies, we also have a similar limitation: the lack of generalizability of results. We advise that the effect of the same methodology and technique should be further investigated in the form of case series and experimental studies. We could not use structured assessments, such as scales or questionnaires, to identify this patient’s functional and quality of life composite scores. Hence, future studies should consider incorporating more objective evaluations for assessing the function and participation of the patients.

## 4. Conclusions

After three months of follow-up, we could see that the patient is walking independently and can raise their leg in the air. Currently, after two and half years of injury, the patient is walking independently and has had no re-injuries. There were no complications reported after this surgery. Hence, we assume that this rare condition, like bilateral quadriceps rupture post-total knee replacement, can be surgically repaired directly with the help of the Kessler technique using simple fiber tape without any grafts or mesh materials. 

## Figures and Tables

**Figure 1 healthcare-11-00631-f001:**
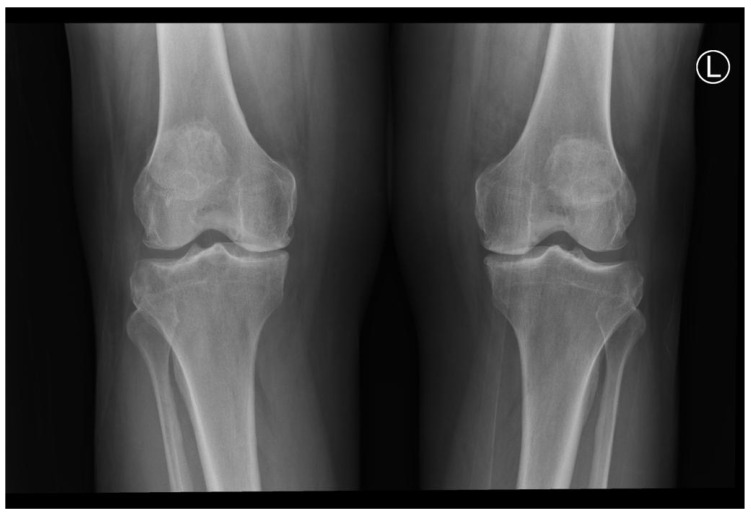
Patient’s X-rays showing osteoarthritic knee changes.

**Figure 2 healthcare-11-00631-f002:**
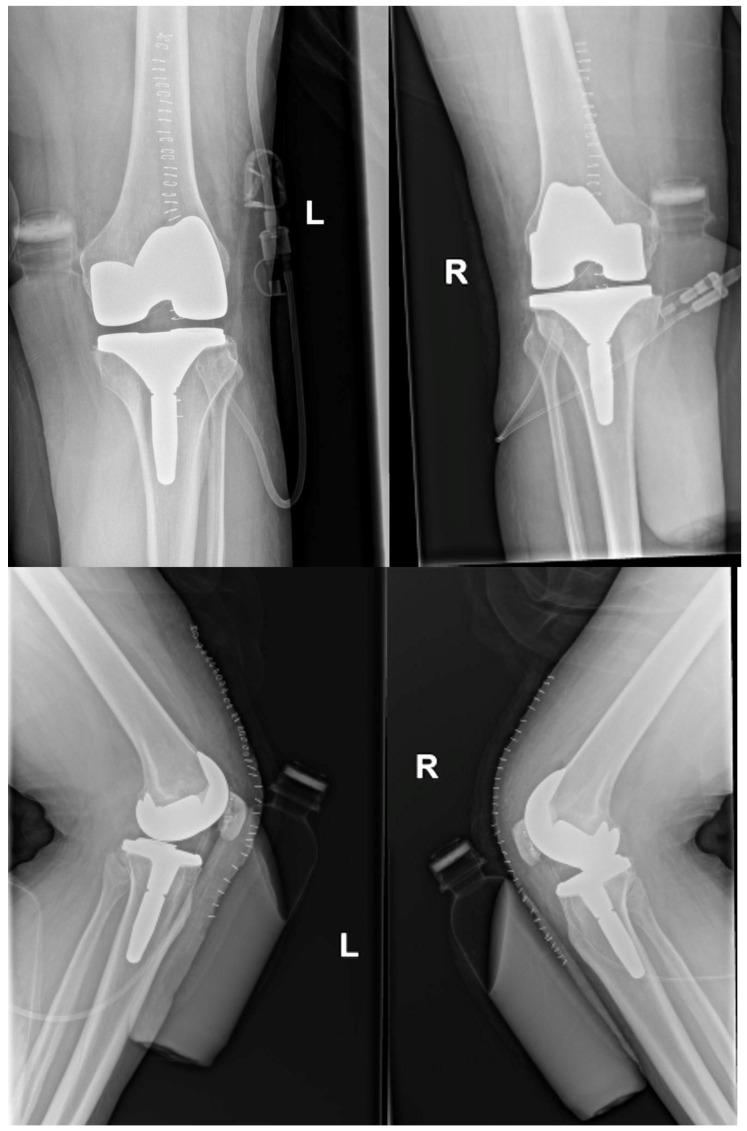
Showing postoperative knee joint in anterior-posterior and lateral views.

**Figure 3 healthcare-11-00631-f003:**
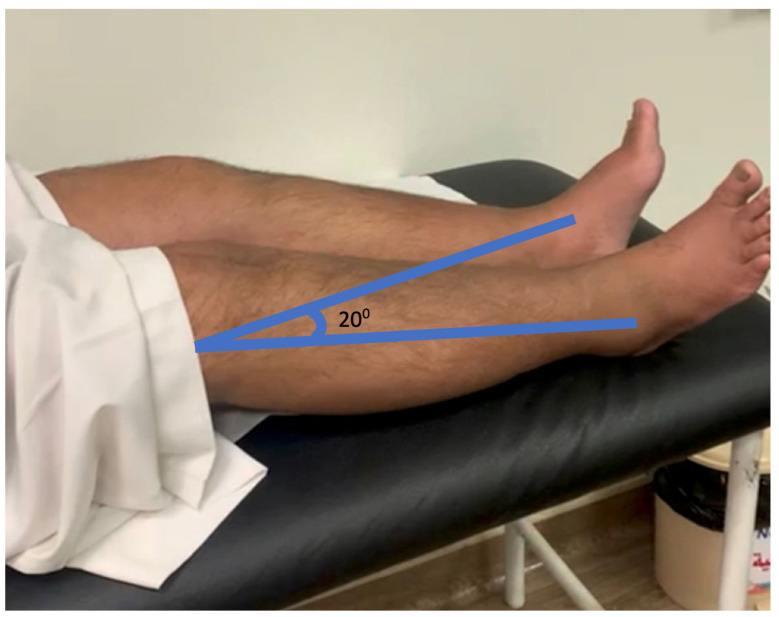
The patient showing disruptive extensor mechanism by right positive leg-raising test. The patient is trying to lift the right leg; the thigh portion is lifting from the bed surface, but the heel is not lifting due to a complete rupture in the quadriceps muscle.

**Figure 4 healthcare-11-00631-f004:**
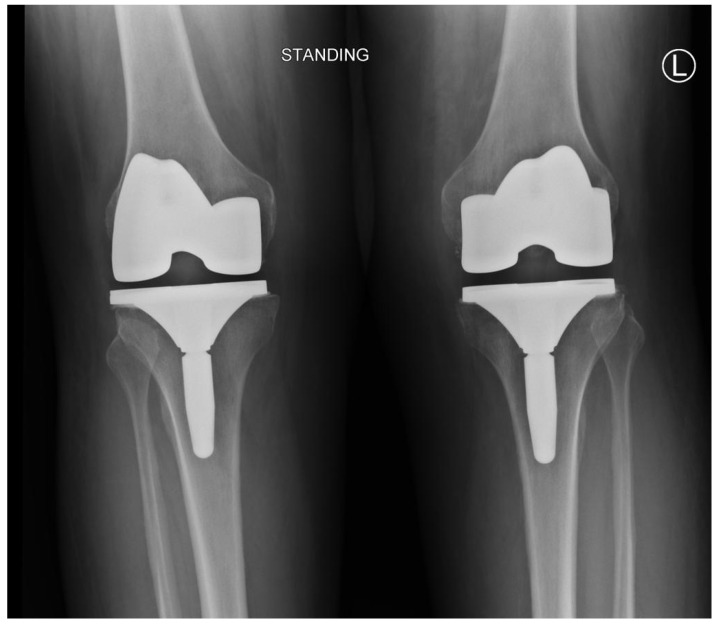
The patient’s standing bilateral knee X-ray did not reveal any periprosthetic fracture.

**Figure 5 healthcare-11-00631-f005:**
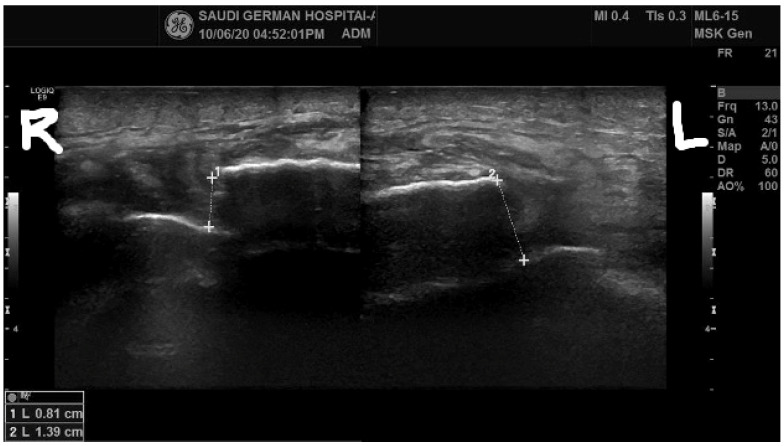
Ultrasound report of the quadriceps tendon with complete bilateral rupture. R: Indicates right side; L: Indicates left side.

**Figure 6 healthcare-11-00631-f006:**
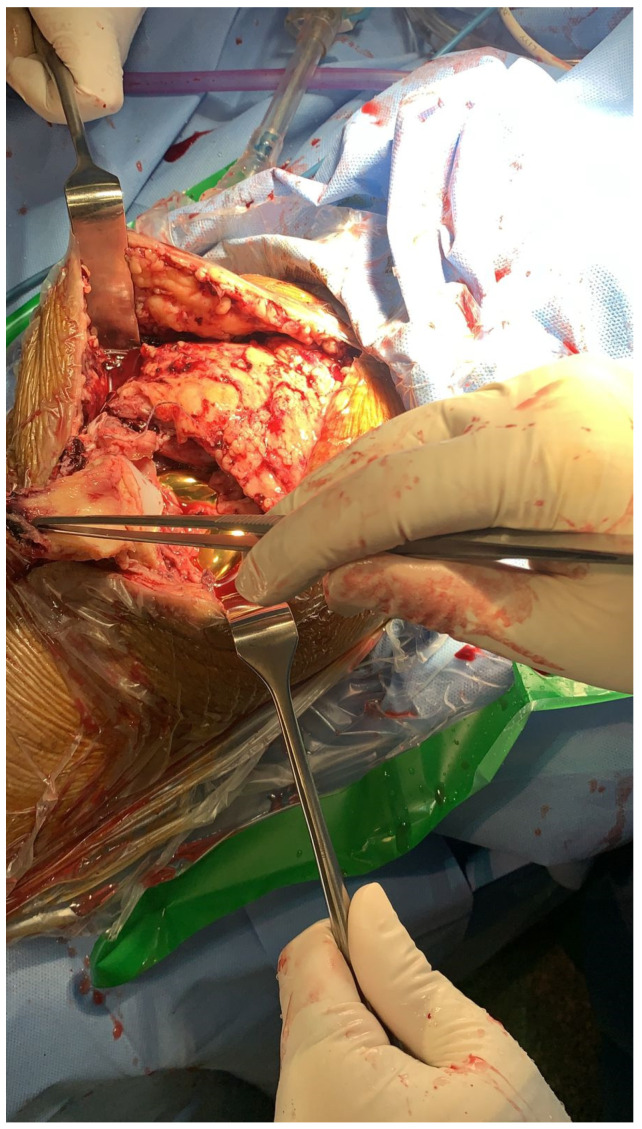
Demonstrating quadriceps rupture intraoperatively.

**Figure 7 healthcare-11-00631-f007:**
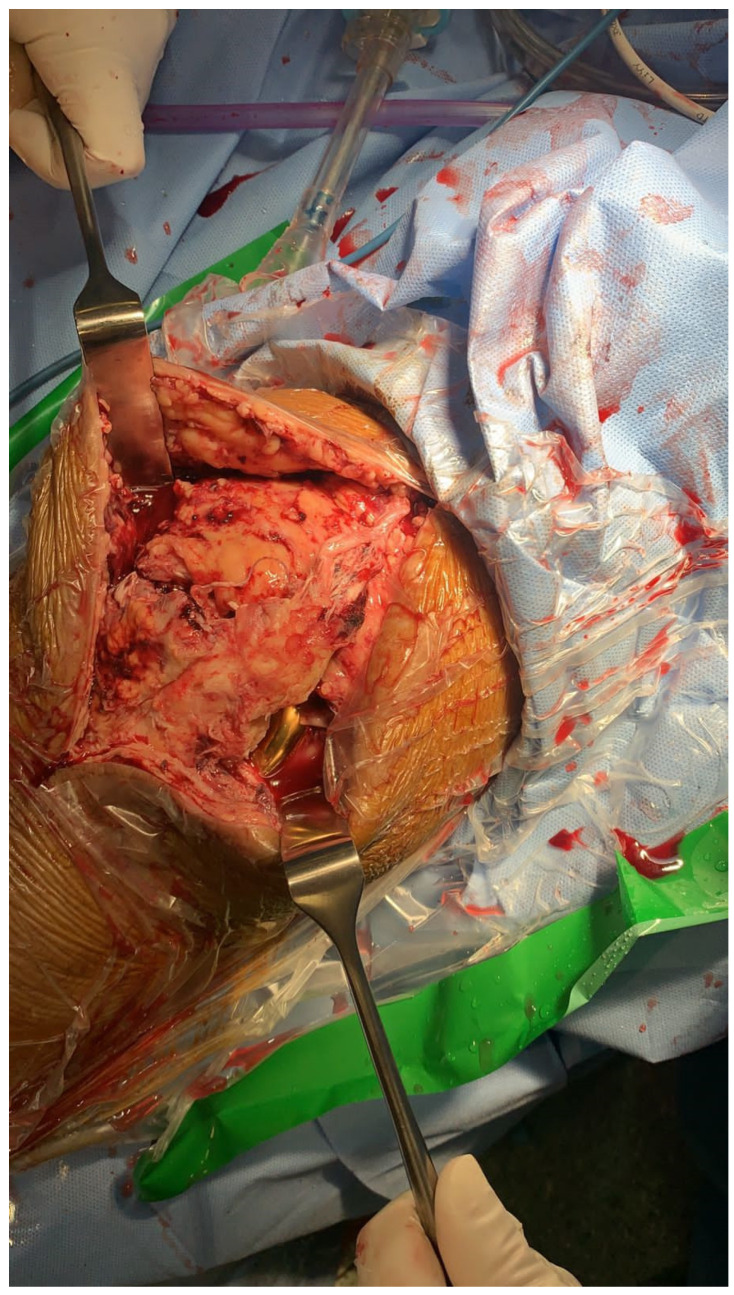
Representing the intra-operative repair of the quadriceps muscle.

**Figure 8 healthcare-11-00631-f008:**
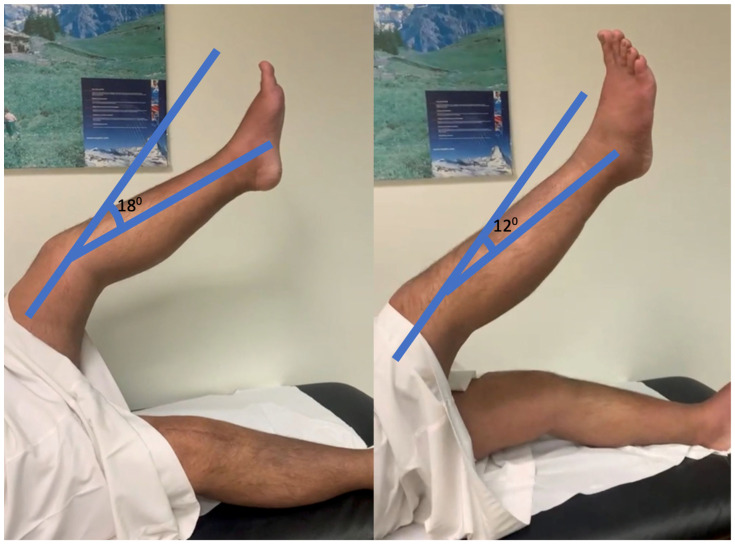
Showing the active hip knee extension in the left and right lower extremity.

**Table 1 healthcare-11-00631-t001:** Training details of the high sitting quadriceps and hamstrings resistance exercise.

Resistance Training Characteristics	7th Week	8th Week	9th Week	10th Week	11th Week Re-Check 1RM and Assign a New Load	12th Week
Load percentages of 1RM	40	50	60	70	50	60
Number of repetitions	6–8	8–10	10–12	12–14	8–10	10–12
Number of sets *	3	3	3	3	3	3
Number of sessions per week	4	5	5	5	4	5

Note: 1RM: One repetition maximum. * 5–10 s rest between sets.

## Data Availability

Data is available with the corresponding author mentioned in this research paper.

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
