# Peer review of "Management of Bilateral Quadriceps Tendon Ruptures Post Total Knee Arthroplasty by Kesler Technique Using Fiber Tape"

_healthcare, 2023, doi:10.3390/healthcare11050631_

Round 1
Reviewer 1 Report
Thank you for the opportunity to review the paper.
Very interesting topic for the scientific community, however, lacks some structure in its presentation.Abstract
I suggest that the key words are not the same as those already in the title, to promote a greater dissemination of the paper
Introduction
Already in the introduction you should start introducing the case and it is expected that you end with an objective of this case study.
Case Description
The purpose of figure 3 and 8 is not perceptible. Is it really necessary to have this image? There may even be differences, but this method of registration is very subjective.
This paper should be more clearly structured, with a section for the intervention, showing the before and after.
It is important to mention the ethical considerations of the study.
It is also important to have a section with the results.
The discussion presented is very poor considering the case study, it would be pertinent to also highlight the relevance of this surgical technique.
There is a lack of response to the proposed objective.
They should review the way they cite references throughout the document.
Author Response
Thank you so much for your valuable time reviewing and considering our study. The suggestions offered by you were constructive. We carefully thought of changes in the manuscript as per your suggestions. We modified and highlighted the changes in the manuscript. The highlighted changes are in yellow for reviewer-1 comments. Here is a point-by-point explanation of your comments. Thank you once again,

Reviewer 2 Report
The presented work is a limited study and carried out on a single patient so its impact and significance is yet to be determined.
There are a few grammatical mistakes that need to be addressed. The reference number at the end of a sentence should be before the full stop. There are 24 references but only 22 are cited in the text. The page numbers of most of references are wrong. In Figure 5, presenting should be removed from the caption. Also, what is the significance of ultra sound test. Can it be replaced by MRI. There are a few VR Physiotherapy exercises and in future, a few of them can be recommended.
Author Response
Respected Editor and Reviewers,
Thank you so much for your valuable time reviewing and considering our study. The suggestions offered by you were constructive. We carefully thought of changes in the manuscript as per your suggestions. We modified and highlighted the changes in the manuscript. The highlighted changes are in green for reviewer-2 comments. Here is a point-by-point explanation of your comments. Thank you once again,

Reviewer 3 Report
Dear authors
You have written an interesting case report.
Initially, please correct all references according to journal instructions. The references must be at the sentences' end before the dot.
In the introduction, there is no rationale or information on the Kesler technique and why this was a preferred technique in your case. Please add relevant literature.
The last sentence of the introduction (lines 58-60 ) needs to be backed up by references.
The paragraph in lines 89-93 and Figure 5 - In the figure, the measurements are 1.39 cm and 0.81 cm. However, you report 2.4cm on the right side. Please adjust the Figure so that the right values can be seen in the Figure.
Line 115 - did the patient have any isometrical exercises or electrical stimulation prescribed during the 6-week immobilisation?
Lines 118 to 121. This section needs to be more specific to allow repeatability. How did you measure repetition maximum, which exercise did you do, report sets, reps, progression? I recommend adding a table of exercises and progression.
Line 126- single leg raisin / reference needed for this test - add
Line 145 - correct reference place.
Can you report the current patient status - is he still uninjured - report the total time after the injury, or have there been any complications?
Were there any limitations to your procedure or data,...? A limitations paragraph needs to be added at the end of the discussion.
Overall an interesting case study that needs some further work. Therefore, I recommend a major revision.
Kind regards
Author Response
Respected Editor and Reviewers,
Thank you so much for your valuable time reviewing and considering our study. The suggestions offered by you were constructive. We carefully thought of changes in the manuscript as per your suggestions. We modified and highlighted the changes in the manuscript. The highlighted changes are in blue for reviewer-3 comments. Here is a point-by-point explanation of your comments. Thank you once again,

Round 2
Reviewer 1 Report
Thank you for the opportunity to review the paper again.
I agree with most of the changes and would like to congratulate you on the way you have rewritten the whole paper. It has become much clearer.
At this moment I have only one suggestion to be carried out In figures 3 and 8 it would be pertinent to place/draw the angle at the knee joint in substitution of the horizontal line.
Author Response
Thankyou so much for your efforts and patience with us. Because of your eagle-eye view, we are able to make our manuscript better. As per your suggestion, we have modified figures three and eight. Kindly go through them and accept our manuscript. Thank you once again.
Reviewer 3 Report
Dear Authors,
Thank you for addressing my questions and suggestions.
There is some more information that needs to be added to your case study for greater repeatability.
In the Table 1 - Training details - please add the duration of break/pause between the exercises and sets. Please just state repetitions, sets and sessions in the table's first column and not repetitions/sets as it is confusing.
Additionally, the Conclusion subheading is missing. Please add it and restructure your text - The limitation of the study is the last paragraph of the discussion.
Overall, I recommend a minor revision.
Kind regards
Author Response
Thank you so much for your patience with us. We modified the table as per your instructions on line 160 and added a conclusion heading on line 222. Your extremely helpful comments by you, and because of your suggestions, our paper became better and more understandable. Kindly accept our paper now. Thank you